# Research Progress on Laser Cladding Alloying and Composite Processing of Steel Materials

Tengfei Han [1,2,3], Kexin Zhou [2,3], Zhongyu Chen [1] and Yuesheng Gao [4,*]

1   College of Mechanical Engineering, Nanjing Vocational University of Industry Technology (NJUIT), Nanjing 210023, China
2   State-Owned Wuhu Machinery Factory, Wuhu 241007, China
3   College of Materials Science and Technology, Nanjing University of Aeronautics and Astronautics (NUAA), Nanjing 210016, China
4   Arkema-ArrMaz, R&D Mining Lab, Mulberry, FL 33860, USA
*   Correspondence: ygao4@mtu.edu

**Abstract:** Laser cladding technology is a reliable and efficient surface modification technology, which has been widely used in surface alloying and composite processing of steel materials. Firstly, the characteristics of laser cladding technology were introduced, and the effects of process control and the material system on the geometric shape, size, microstructure, and properties of cladding coating were analyzed by summarizing the research results of laser cladding on steel surfaces. The results show that with the increase of laser power, the dilution rate and width of the cladding coating increase, and the grain becomes coarse. Thus, the wear resistance deteriorates. Compared with alloy cladding coating, composite cladding coating exhibits better wear and corrosion resistance, but the plastic toughness is worse than alloy cladding coating. The research progress of surface alloying and composite processing of steel worldwide was analyzed from various aspects. Current results suggest that laser cladding alloying and compounding can enhance the wear resistance and corrosion resistance of steel materials. Based on the summary of the current research results, the development prospect and planning of laser cladding technology in the field of surface alloying and composite processing of steel are further pointed out.

**Keywords:** steel materials; laser cladding; alloying; compound; research progress

## 1. Research Background

Due to the unique processing and mechanical properties of steel materials, they are widely used in the fields of vehicles, ships, ocean engineering, chemical equipment, and aerospace [1–3]. In many industries with poor working conditions, mechanical equipment is often subjected to periodic or continuous abrasive wear, erosion wear, and composite wear, which causes surface wear failure of parts and components of mechanical equipment and reduces the service life of mechanical equipment. Wear, corrosion, and fatigue are the three main failure modes of materials [4–6]. It has been reported that 70–80% of mechanical equipment failures are due to wear and tear of all kinds [7,8]. In addition, with the proposed implementation of the "Made in China 2025" development strategy, higher requirements have been put forward for the quality and efficiency of mechanical equipment manufacturing.

In 1976, Gnamuthu D.S obtained a patent for the laser cladding of one metal coating to another metal substrate. Laser cladding technology has been important to the field of material surface strengthening since its discovery. Laser cladding is a method of adding powder materials or wire materials on the surface of the substrate and employing a high-energy density laser beam to melt materials together with the thin layer on the surface of the substrate, forming a metallurgical composite cladding coating. If the cladding material is wire, wire feeding equipment is usually used to transport the wire to the surface for

cladding. Theoretically, the wire cladding material's utilization rate can reach 100%. If the cladding material is powder, laser cladding can be divided into two types: powder laying and powder feeding, according to the different ways of material supply. Powder laying laser cladding is easy to operate and does not need special powder feeding equipment, but the powder utilization rate and production efficiency are low. Powder feeding laser cladding needs special powder feeding equipment to transport the powder to the surface to be fused, and the powder utilization rate and production efficiency are higher.

Laser cladding has been a widely used surface modification technology. The cladding coating prepared by this technology has excellent performance, and the thickness of the cladding coating can be changed from a few microns to a few millimeters. The thickness of the cladding coating can be flexibly adjusted according to the actual user demand, which also considerably saves the cladding materials. Therefore, laser cladding has become an important means to improve parts' surface wear and corrosion resistance. In addition, laser cladding technology can save rare and precious metals, reduce energy consumption, in line with the national energy conservation, emission reduction, green and low-carbon strategic development goals.

## 2. Laser Cladding Process of Steel

Laser cladding is an instantaneous processing process. Although the cladding process is very short, it contains rich interactions of the laser with the cladding material and substrate material [9,10], the complex metallurgical physicochemical reactions after the formation of the molten pool, and the rapid solidification process of the molten pool [11,12]. The completion of each stage of the cladding process is closely related to the technological parameters of laser cladding. The main technical parameters of laser cladding include laser power, scanning speed, and laser spot diameter. The dilution rate is one of the important parameters to characterize the quality of the cladding coating. The dilution of the cladding coating by the substrate material is inevitable [13], and the substrate's surface must be melted to form good metallurgical bonding at the interface between the cladding coating and the substrate. However, to maintain the properties of the substrate material and cladding coating and avoid the influence of substrate dilution, the dilution rate should be controlled in an appropriate range. Each process parameter affects the surface formability, dilution rate, microstructure, and phase composition of the cladding coating [14]. It is necessary to optimize the process parameters and control them within a reasonable range to obtain the cladding coating with good surface molding and a dense internal microstructure.

The laser is the heat source of the cladding process. Laser power is directly related to the heat input in the cladding process. When the laser power is too large, the surface roughness of the cladding coating increases due to the increase of disturbances in the molten pool, and the molten pool easily absorbs the gas. The excessive heat transfer to the substrate leads to a rise in the dilution rate, which affects the cladding coating's surface shape and chemical composition [15–17]. On the contrary, when the laser power is too small, the cladding material cannot completely melt, and the wettability between the molten cladding materials and the substrate becomes worse, which leads to poor molding of the cladding coating. The laser energy absorbed by the substrate surface is insufficient, and the substrate cannot form a good metallurgical combination with the cladding coating.

### 2.1. Laser Power

Laser power has a direct effect on the cladding coating. The influence of laser power on the coating is manifold. Zhang et al. [18] found that the volume fraction of the amorphous phase in the cladding coating reached 92% at 4 kW for a cobalt-based composite coating. With the increase of the laser power, the average hardness of the cladding coating decreased, but the wear resistance of the cladding coating improved. Fan et al. [19] found that when the laser power is 2 kW, the depth of the molten pool is 0.14 mm (as shown in Figure 1a), and there may be a weak connection between the cladding coating and the substrate. When the laser power is increased to 3.2 kW, the depth of the molten pool is 0.75 mm (as shown



in Figure 1b), the dilution rate of the cladding coating increases, and the performance decreases. It can be seen from the research results that laser power has a great influence on the width and depth of the coating but has little impact on the height of the coating. The hardness of cladding coatings decreases with the increase of the laser power because the higher the laser power, the higher the dilution rate of cladding coatings, and the lower the performance. Fu Hanguang's team of Beijing University of Technology [20] applied laser cladding to a NiCrBSi cladding coating on a 42CrMo steel surface, and it was found that the dilution rate of the cladding coating was 5.9% when the laser power was 1500 W and 48.7% when the laser power was increased to 3500 W. The penetration depth and dilution rate of the substrate increase with the increase of laser power, and the hardness of the coating decreases with the increase of the laser power.

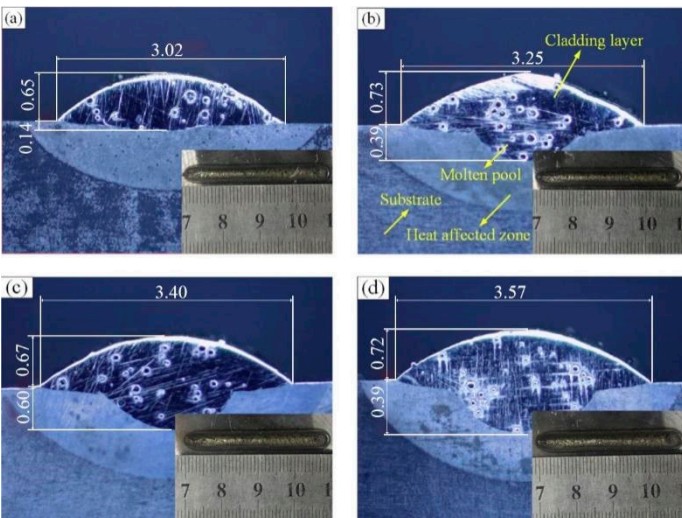

**Figure 1.** Macroscopic structure of a single-channel cladding coating with different laser powers: (**a**) 2.0 kW, (**b**) 2.4 kW, (**c**) 2.8 kW, and (**d**) 3.2 kW (measure unit is mm).

## 2.2. Laser Cladding Speed

The effect of laser cladding speed on the quality of the cladding coating is similar to that of laser power [21]. If the cladding speed is too high, the cladding powder cannot be fully melted [22], while if the cladding speed is too low, the powder material will be burned out as the laser irradiation time is too long [23].

Dara et al. [24] focused on laser cladding an AISI 316L stainless-steel cladding coating with feeding powder, and it was found that the cladding speed accounted for 59% of the total factors of cladding coating height variation, and the powder feeding rate accounted for 30% (as shown in Figure 2a). With the increase of the cladding speed, the height of the cladding coating decreases. The rate of laser cladding mainly affects the amount of powder delivery. As the speed of cladding increases, the amount of powder delivery decreases. As the powder feeding rate increases, the cladding coating height also increases. The cladding speed is the most important factor for the height variation of the cladding coating. As shown in Figure 2b, laser power accounts for 58% of the total factors for the interpretation of the cladding coating width, which is the most important factor. It needs to be emphasized that cladding speed accounts for 24% of the total variation of the cladding coating width, which is the second factor affecting the width of the cladding coating. With the increase in cladding speed, the width of the cladding coating becomes narrower. As shown in Figure 2c, the cladding speed and powder feeding rate account for 60% and 23% of the total factors of wetting angle variation between the cladding coating and the substrate, respectively, while the laser power only accounts for 2%, indicating that the cladding speed and powder feeding rate are the controlling factors of cladding coating geometry. Moreover, the powder feeding rate depends on the cladding speed. It can be seen from the results

that the laser cladding speed (scanning speed) is the main factor affecting the height and wetting angle of the cladding coating.

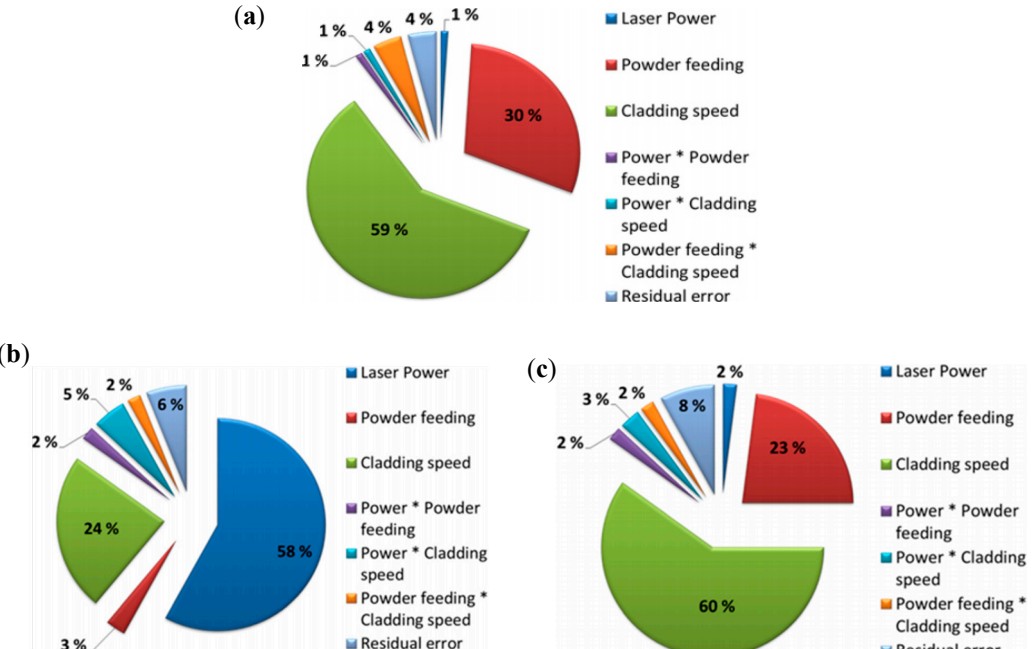

**Figure 2.** Influence of laser cladding process parameters on the geometric shape of the cladding coating. (**a**) The effect of cladding speed on the height of the cladding coating, (**b**) the effect of laser power on the change of the cladding coating width, and (**c**) the effect of cladding speed on the change of the wetting angle between the cladding coating and the substrate.

### 2.3. Laser Beam Diameter

At present, the laser beam is usually circular, and the energy distribution of the laser beam often follows the Gaussian distribution [25]. When other process parameters are fixed, if the beam diameter is too small, it is difficult to obtain a large area of cladding coating, which affects the efficiency of laser cladding. If the beam diameter is too large, the laser beam heat is easy to disperse, which affects the quality of the coating. When the laser heat input energy is constant, the thickness of the prepared powder material is too large; thus, the laser cannot melt the powder bed and cannot form an effective cladding coating. Therefore, the amount of molten powder and laser beam diameter also have a certain relationship.

Kong et al. [26] studied and compared austenitic stainless-steel parts with three beam diameters (0.75 mm, 1.5 mm, and 3 mm) from four aspects: microstructure, mechanical properties, powder utilization, and finished surface quality. The results show that when the beam diameter is smaller, the grain size is finer, the ferrite content is lower, and the dislocation density is higher, which makes the forming parts have better mechanical properties because of the high cooling rate of the molten pool. On the contrary, when the beam diameter is larger, the molten pool is more stable, the powder splash is less, the material utilization is higher, and the surface quality is better.

### 2.4. Supply of Cladding Materials

There are two kinds of laser cladding powder feeding methods, that is, powder preparation and powder feeding. When the thickness of the powder material is small, a higher laser energy is required to be transmitted to the substrate, which increases the dilution rate of the cladding coating and hurts the performance. Similar to the influence of the preset powder material thickness, the powder feeding rate also directly affects the interaction between the laser and the powder material, thus affecting the quality of the cladding coating. The supply of cladding materials has a great impact on the

surface-forming quality of the cladding coating, the internal structure, and the internal quality defects.

The parameters of laser cladding technology have significant effects on the dilution rate of the substrate [27], the surface-forming quality [28], the geometric size [29], and the mechanical properties of the coating [30]. The influence weight of each process parameter on a certain coating property is also different. The different cladding materials and laser types will change the influence of laser cladding process parameters on the coating. Moreover, it is found that the technological parameters do not affect the quality of the cladding coating independently, but they affect each other. Therefore, when selecting the technical parameters of cladding, based on considering the main influencing factors and cladding material system, the synergistic effect between the technical parameters of cladding should also be considered. Now, the research and optimization of laser cladding technology have become an important topic in the theoretical investigation of laser cladding.

## 3. Laser Cladding Materials of Steel

Laser cladding materials directly affect the chemical composition of the coating. Therefore, the choice of cladding materials has an important influence on the quality and performance of the coating. Currently, most commercial alloy powders are designed and produced according to the technological characteristics of flame spraying, plasma spray welding, and powder metallurgy, which cannot meet the specialized needs of laser cladding. Although laser cladding materials apply to a wide range of substrates, there are some more suitable for a certain cladding substrate in a certain working environment. Selecting appropriate cladding materials is a prerequisite for obtaining good surface-forming quality, a smooth surface, and an internal defect-free coating for the determined substrate material. According to the technological characteristics of laser cladding, generally, the selection of laser cladding materials follows the following principles [31].

(1)　The close thermal expansion coefficient principle between cladding materials and the substrate material is provided as follows. Laser cladding is a process in which the cladding materials melt rapidly under laser irradiation and then cool sharply and solidify. Due to the rapid heat and cold, there will be a certain thermal stress between the cladding coating and the substrate. Suppose the thermal expansion coefficient of the cladding materials and the substrate is different. In that case, the thermal cracks between the cladding coating and the substrate are easy to occur under thermal stress, which will affect the performance of the cladding coating. On the contrary, if the thermal expansion coefficient of the cladding materials and the substrate is close, it is beneficial to reduce the thermal stress between the cladding coating and the substrate and reduce the tendency of cracks in the cladding coating.

The matching principle of the linear expansion coefficient between laser cladding materials and the substrate material is as follows:

$$-\sigma_2(1-\gamma)/(E\Delta T)\langle\Delta\alpha\langle\sigma_1(1-\gamma)/(E\Delta T) \tag{1}$$

where $\sigma_1$ is the tensile strength of the cladding coating (MPa), $\sigma_2$ is the tensile strength of the substrate (MPa), $\Delta\alpha$ is the linear expansion coefficient difference between the cladding coating and the substrate, $\Delta T$ is the difference between the cladding temperature and room temperature, $E$ is the elastic modulus of the cladding coating, and $\gamma$ is Poisson's ratio.

After laser cladding, the main reason for cracking is residual stress in the cladding coating. Thermal stress, phase transition stress, and constrained stress are the main components of residual stress, among which thermal stress accounts for the heaviest proportion compared with the other two. The thermal stress of laser cladding is determined as follows [32]:

$$\sigma_{\text{th}} = E\Delta\alpha\Delta T/(1-\gamma) \tag{2}$$

where $\sigma_{\text{th}}$ is thermal stress, $E$ is the elastic modulus of the cladding coating, $\gamma$ is Poisson's ratio, $\Delta\alpha$ is the difference of the linear expansion coefficient between the cladding coat-

ing and the substrate, and $\Delta T$ is the difference between the cladding temperature and room temperature.

From Equation (2), it can be seen that when the difference between the linear expansion coefficient of the cladding coating and the base material is smaller (in other words, when the linear expansion coefficients of the cladding coating and the base material are closer), the thermal stress of $\sigma_{th}$ becomes smaller. Then, the cracking tendency of the cladding coating is the smallest.

(2) The second principle is based on similar melting points between cladding materials and substrate materials. After melting the cladding materials and the surface materials of the substrate, the molten pool is formed, and the molten pool solidifies to form the cladding coating. The melting point of the cladding materials and the base material is similar, which can ensure that the cladding materials and the surface material of the substrate melt simultaneously and form a metallurgical combination to reduce the dilution rate of the cladding coating. When the melting point of the cladding materials is much lower than the melting point of the base material, the surface material of the substrate cannot reach the melting point. It cannot melt after the laser energy melting of the cladding materials, and the substrate cannot form a metallurgical combination with the cladding coating. After increasing the laser heat input, the surface material of the substrate melts. At this time, the melting temperature of the cladding materials to form the molten pool exceeds the melting point of the cladding materials. The cladding materials will be vaporized, resulting in the loss of the cladding materials and easy-to-produce pores, shrinkage holes, and other defects in the cladding coating. When the melting point of the cladding materials is much higher than the melting point of the base material, the cladding materials have not been melted when the base material is melted. This process will cause the burning of the base material, and hence it cannot form a good cladding coating.

(3) The third principle is applying the wettability of cladding materials to substrate materials. After the molten pool is formed, the cladding materials need to spread on the surface of the substrate to form the cladding coating. The good wettability between the cladding materials and the substrate is the prerequisite for the molten pool to spread on the surface of the substrate and is also the key to the good molding performance of the cladding coating [33]. The wettability of cladding materials on the surface of the substrate is directly related to the surface tension of the molten pool. Increasing the temperature of the molten pool is an effective way to reduce the surface tension of the molten pool. Still, a too-high temperature of the molten pool will also cause the burning of alloying elements.

At present, the commonly used cladding materials for laser cladding are metal (alloy) powder [34], ceramic powder [35], and metal-ceramic composite powder [36]. In addition to following the above three principles, selection of cladding materials is made according to requirements such as actual needs and expected performances.

### 3.1. Metal (Alloy) Cladding Materials

Metal cladding materials mainly include iron-based alloy [37], nickel-based alloy [38], and cobalt-based alloy powder [39]. It is noteworthy that with the development of laser cladding technology and the continuous improvement of the requirements for the quality and performance of cladding coatings, high-entropy alloy powder cladding materials [40,41] and self-designed alloy powder cladding materials [42,43] also emerged.

Iron-based alloy powder: At the beginning of laser cladding technology, iron-based alloy was used as a laser cladding material, which is suitable for wear resistance requirements that are not too high and easy to deform parts, ideal for choosing cast-iron and low-carbon steel materials as the substrate material. Its biggest advantage is a wide range of sources of ferrous alloy materials. Due to the similar composition as the base material, the interface of the cladding coating forms, and the substrate is firmly bonded [44]. The cladding coating is mainly composed of a metastable structure, and a martensite structure and carbide are

produced in the cladding coating, which improves the hardness and wear resistance [45]. In addition, the price of the iron-based alloy is 1/4~1/5 of that of nickel-based alloy and 1/8~1/9 of that of cobalt-based alloy, so the cost is low and more suitable for large-scale production [46,47]. The disadvantages are high melting point, poor flow performance of the molten pool, easy oxidation, and easy to produce cracks, pores, slag, and other defects in the cladding coating.

Nickel-based alloy: Laser cladding nickel-based alloy mainly includes its alloy and a nickel-based self-fused alloy. The performance of nickel is second only to cobalt. It can withstand higher temperatures, has a good heat resistance and corrosion resistance [48], and is much cheaper than cobalt. Therefore, nickel-based alloy cladding materials are widely used and dominate in many acid-resistant and heat-resistant working conditions [49]. The alloying mechanisms of nickel-based alloy are: (1) the solid-solution strengthening of austenite by elements such as Cr, Fe, Co, and W [50], (2) the synthesis of intermetallic compounds by elements such as Fe, Ti, Al, and Ni for precipitation strengthening [51,52], and (3) the grain boundary strengthening by adding elements such as B, Zr, and Co. In engineering applications, one or several reinforcement mechanisms are determined according to the actual performance design and application requirements.

Cobalt-based alloy: Cobalt-based alloy cladding materials have particularly good high-temperature performance, wear resistance, and corrosion resistance, and comprehensive mechanical properties, which are suitable for high-temperature resistance, oxidation resistance, wear resistance, corrosion resistance, and thermal fatigue resistance equipment and parts [53]. The main alloying elements used in cobalt-based alloys are Co, Ni, C, W, Cr, and Fe [54–56]. Among them, Ni can reduce the thermal expansion coefficient of the cobalt-based alloy cladding coating, reduce the melting temperature range of the alloy, effectively improve the anti-crack sensitivity of the cladding coating, and improve the wettability of the cladding coating to the substrate [57].

High-entropy alloy: This alloy is formed by five or more elements in nearly equal molar ratios. A solid solution is the main phase composition of high-entropy alloys, so the strengthening form is mainly solid-solution strengthening. Compared with the traditional alloy, its specific strength has been greatly improved, and it has good corrosion resistance, high-temperature oxidation resistance, wear resistance, and good comprehensive mechanical properties. Due to these advantages, high-entropy alloys are also used as laser cladding materials, and this continues to be an active area of research today [58,59]. However, it is found that due to the rapid melting of laser cladding and the difference of diffusivity of each element, there will be certain composition segregations, making the structure of non-high-entropy alloys unable to meet the design [60–62].

Independent design alloy: Compared with mature alloy cladding materials' systems, the independent design of alloy composition is more flexible. There is no fixed choice of elements, and it is not constrained by the existing component design principle. However, according to the use of the environment, the expected performance requirements are in combination with phase diagram elements to select and design the elements' ratio, which is beneficial to the preparation of an excellent performance of the laser cladding coating [63].

### 3.1.1. Ceramic Cladding Materials

Ceramic materials are known for their excellent acid and alkaline resistance, high hardness, wear resistance, thermal stability, low density, and light weight [64,65]. In laser cladding technology, ceramic powders as cladding materials are mainly used in the working environment with severe wear and strong corrosions. However, there are great differences in physical and chemical characteristics between ceramic powder and metal substrates, such as in the linear expansion coefficient, elastic modulus, wettability, melting point, and thermal conductivity [66,67]. Due to the mismatch of physical and chemical properties, the ceramic cladding coating tends to produce pores, cracks, and even spalling due to the insufficient bonding force between the cladding coating and the substrate. This is not conducive to direct laser cladding of ceramic materials on the surface of the metal

substrate to prepare the cladding coating. In addition, due to the large elastic modulus of ceramic materials, the ceramic cladding coating is hard and brittle, has poor plasticity and toughness, and poor fatigue resistance, which cannot bear the impact force well. Under the action of impact force, the cladding coating is prone to brittle fracture damage. Therefore, pure ceramic cladding is not suitable for heavy loads, impact loads, and high-stress working conditions.

3.1.2. Metal-Ceramic Composite Cladding Materials

The metal powder material and metal substrate have good wettability and bonding, but the wear resistance and corrosion resistance are slightly poor. The ceramic powder material has high hardness, good wear resistance, and good corrosion resistance, but the wettability and bonding performance between ceramic and metal substrates are very poor. The metal-ceramic composite powder can give full play to the advantages of metal powder and ceramic powder and make up for the disadvantages of each [68]. Metal-ceramic materials can be divided into coated composite powder and mixed composite powder in hybrid mode [69]. The coated composite powder is a composite powder with a core-shell structure formed by coating the surface of ceramic particles with metal by mechanical alloying or chemical synthesis. The hybrid composite powder is the mixed powder obtained by automatically mixing metal powder and ceramic powder. Ceramic and metal powders have no combination of physics and chemistry, and are easier to prepare than coated powder. The excellent performance of metal-ceramic composite powder makes it the most widely used in laser cladding [70,71].

**4. Laser Cladding Alloying and Compounding of Steel Materials**

*4.1. Laser Cladding Alloying of Steel Materials*

Laser cladding alloy and steel share similar physical and chemical properties compared with nonmetal material, which can improve steel's material surface properties and exhibit good wettability between the alloy coating and the steel substrate [72,73]. This is advantageous to the alloy coating laser melting compound preparation.

Lin et al. [74] found that with the increase in Mo content, the Mo-rich phase of the mesh structure increased, and the grain size of the cladding coating decreased (as shown in Figure 3). After testing the hardness and wear resistance of the coating, it was found that the increase of Mo content can improve the strength of NiAl coating and the toughness and wear resistance of the coating. Wang et al. [75] studied the effect of Co content on the microstructure, composition, hardness, and wear resistance of laser cladding coating on 42CrMo steel. With the increase of Co content, the microhardness of the cladding decreases, but the wear resistance increases gradually. When the Co content is 30 wt.%, the wear resistance of the cladding layer is 3.6 times that of the coating without Co. With the increase of Co content, the wear of the cladding layer changes from abrasive wear to adhesive wear. Gao et al. [76] found that with the increase of Fe content, the phase of the coating changed, and the distribution of Cr became more uniform in the coating. With the increase of Fe content, the corrosion resistance of the coating decreases first and then increases. The corrosion resistance of the coating is the best when the Fe content is 25%. Feng et al. [77] found that Al can promote the formation of the BCC phase in laser cladding nickel-based alloy coatings. The $Al_2O_3$ formed by the Al element will improve the corrosion resistance of the coating, but the excessive Al will reduce the content of $Cr_2O_3$ and thus reduce the corrosion resistance of the coating. In addition, the increase of Al will improve the resistance to plastic deformation of the coating, thereby improving the hardness and wear resistance of the coatings. Yang et al. [78] researched the effect of Cr content on the microstructure and oxidation resistance of laser cladding alloy coatings. The results show that when Cr content is lower than 5 wt.%, the surface of the coating is smooth without cracking. When the content of Cr continues to increase, cracks appear on the coating surface. Adding Cr can promote the formation of a secondary phase and improve the high-temperature oxidation resistance of the coating. Zhang et al. [79] found

that the microstructure of laser cladding FeNiCoCrTi$_{0.5}$Nbx coatings changed with different Nb contents. Nb can afford high coating hardness and wear resistance. The hardness of the FeNiCoCrTi0.5Nb0.5 coating is 2.9 times that of 45 steel.

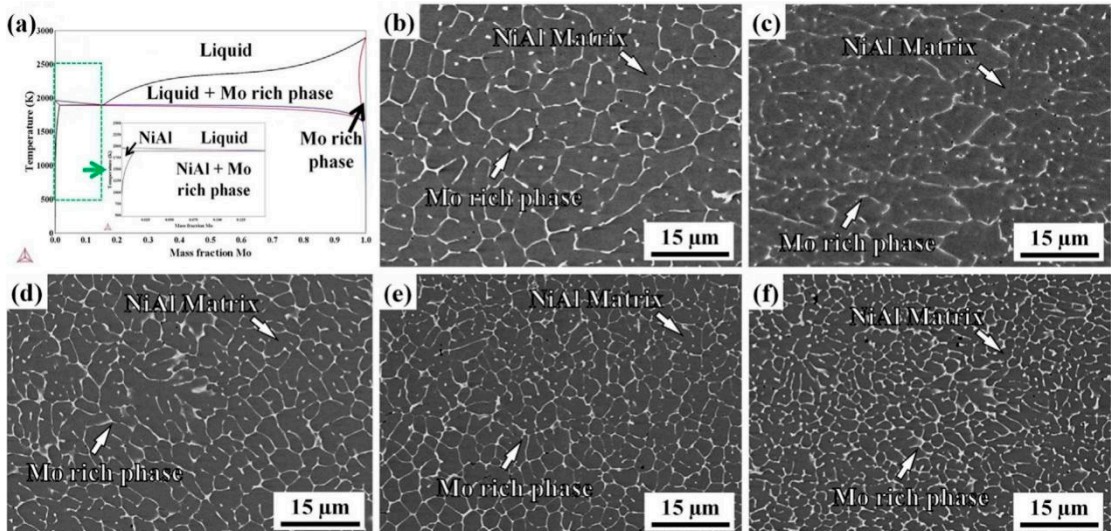

**Figure 3.** (**a**) Simulated phase diagram of NiAl with different amounts of Mo addition. Backscatter SEM micrographs of NiAl coatings with: (**b**) 3 wt.%, (**c**) 6 wt.%, (**d**) 9 wt.%, (**e**) 12 wt.%, and (**f**) 15 wt.% Mo addition.

Wu et al. [80] optimized the process parameters of laser cladding Ni30A coating. It is found that the feeding rate is the main reason for the formation of pores in the coating. Before laser cladding, the cracks in the coating can be eliminated by preheating the substrate to 300 °C in advance and holding it. In this case, the hardness of the coating is 2.7 times that of the substrate. Gao et al. [81] studied the laser cladding FeCrNi coating on the surface of low-carbon steel with different process parameters. At higher scanning speeds, the cooling rate of the mobile solid–liquid interface tends to decrease near the top of the coating. The maximum hardness of the coating was 597 HV$_{0.2}$, which was significantly improved compared with the low-carbon steel substrate (160–220 HV$_{0.2}$).

Zanzarin et al. [82] studied the dilution rate of laser cladding Co alloy powder. It is found that the concentration of iron in the coating increases with the increase of laser-specific energy, indicating that the dilution rate rises. The research team built a model to predict the relationship between specific energy and dilution rates, as represented by predicted values of iron content, as shown in Equation (3):

$$Fe_{predicted} = 0.93 - a \cdot [1 - \exp(b \cdot Es)] \tag{3}$$

where $Fe_{predicted}$ is the predicted value of the iron concentration, $a$ and $b$ are the fitting parameters, and $Es$ is the specific energy.

Murkute et al. [83] prepared a 316L stainless-steel coating on the surface of 1018 steel. The effects of process parameters on microstructure and coating properties were investigated. When the laser energy density, E, is 104.52 J/mm$^3$, the energy is too low to connect different metals, causing massive spherification, geometric warping, and coating delamination. It is also found that the Cr is lost by evaporation during laser cladding. The decrease in laser scanning speed will reduce the hardness of the coating. Tanigawa et al. [84] studied the effect of powder particle size on the heat-affected zone of the laser cladding Ni-Cr-Si-B alloy coating. They found that the smaller the powder particle size, the smaller the heat input required to form the coating, and therefore, the smaller the heat-affected zone. Cracks were found in the cross-sectional edges of the coatings obtained by powder laser cladding with a specific energy of 162.5 J/cm and particle size of 30 μm. Ivan Erdakov et al. [85] studied the effect of the deposition trajectory on the microstructure and mechanical prop-

erties of Ti6Al4V alloy powder materials synthesized by direct laser. The laying method of the track and the direction of the compression test load applied relative to the track position caused the ultimate strength of the Ti-6Al-4V alloy to vary from 1794 to 1910 MPa. The plasticity of the Ti-6Al-4V alloy obtained by direct laser alloying varied from 21.3% to 33.0% depending on the direction of the track laying and the compression test. It should be emphasized that the Ti–Al interface behavior will affect the plastic toughness of the cladding coating [86]. To avoid and eliminate cracks on the workpiece surface, Li et al. [87] studied the effect of process parameters on the evolution of laser cladding. They analyzed the influence of laser power, beam radius, scanning speed, and other process parameters on the changes in the temperature field, flow field, and stress field during the cladding process, which provided an effective way to reduce residual stress. Francisco et al. [88] studied the wear resistance of the laser cladding Ni-Cr-B-Si coating. Figure 4 shows the cross-section morphology of the coating. When the laser heat input is 210.0 J/cm, there is an unconnected region between the C1 coating and the substrate. With the decrease of the laser heat input, there is no unconnected region between the coating and the substrate in C2 and C3. The experimental results show that the coatings prepared by different laser heat inputs have different dilution rates and different wear resistances. Jonas Kimme et al. [89] studied the parameter variation under different preheating temperatures. The experimental results show that the appropriate process parameters and the inductive preheating temperature of 200 °C can produce coatings without cracks and holes on PM steel. Defects such as cracks and pores can be improved by optimizing the process parameters [90]. External energy fields (such as magnetic fields) are helpful in improving cracks and porosity defects [91].

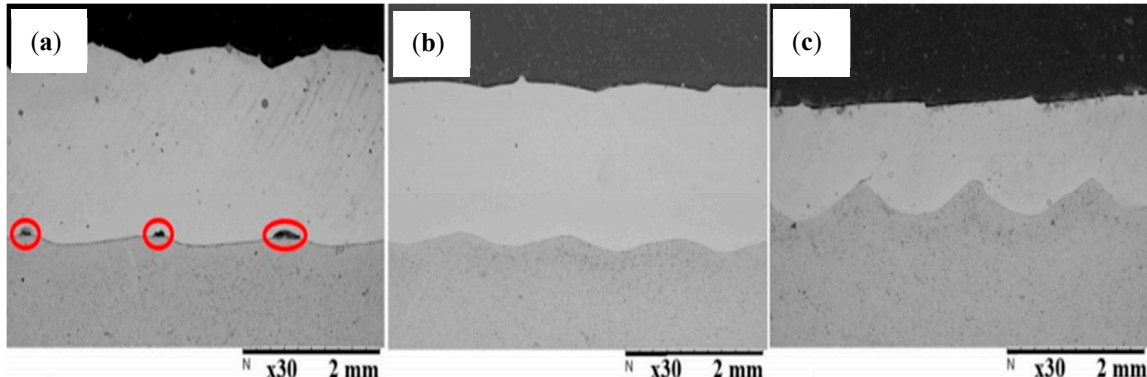

**Figure 4.** Cross-section morphology of Ni-Cr-B-Si coating: (**a**) 210.0 J/cm cross-section morphology, (**b**) 64.6 J/cm cross-section morphology, and (**c**) 58.3 J/cm cross-section morphology.

Researchers have performed comprehensive research on steel surface alloying, mainly focusing on process parameters, dilution rate, microstructure, and mechanical properties (such as hardness, wear resistance, etc.). The relationships among process parameters, microstructure, and properties were analyzed and discussed. The preliminary research laid a foundation for the following surface alloying of carbon steel, but it was found that in the laser alloying process, the metallurgical reaction and reaction mechanism of cladding materials in the laser processing process need to be further studied. Moreover, there are few studies on the effect of a certain element in cladding materials on laser alloying and its mechanism, which need to be further studied.

### 4.2. Laser Cladding Compounding of Steel Materials

Laser cladding on the surface of the steel is based on alloying, adding reinforcing particles to the alloy material. The composite coating not only has good wettability of the alloy coating and the steel substrate but also the mechanical properties and corrosion resistance of the composite coating have been further improved [92,93].

Shen et al. [94] studied the effect of WC content on the crack behavior of laser cladding ceramic-metal coatings. When the WC content is 50 wt.% and 60 wt.%, cracks appear

in the composite coating, while there is no crack when the WC content is 40 wt.%. The crack is mainly caused by the low toughness and high tensile stress of the coatings with high WC content. Hu et al. [95] researched the effect of WC content on the microstructure and properties of nickel-based coatings. The WC-reinforced particles do not affect the enhancement effect, even if they are broken. The hardness and wear resistance of the coating gradually improved with the increase of WC content. When the WC content is 30–60%, the wear resistance of the coating has no obvious change. Qin et al. [96] investigated the effect of different ranges of $Al_2O_3$ on the oxidation behavior of the CoCrAlYTa coating at high temperatures. Adding $Al_2O_3$ particles to the CoCrAlYTa coating increases Al's diffusion coefficient, reduces Al's critical concentration, and promotes the formation of continuous intact $Al_2O_3$ scales. When the $Al_2O_3$ content is 2 wt.%, the CoCrAlYTa coating is oxidized at high temperatures to form a compact scale.

Dariusz Bartkowski et al. [97] investigated the effects of laser cladding parameters on the microstructure, microhardness, chemical Fe-B composition, wear, and corrosion resistance of the composite coatings. With the change of laser cladding process parameters, the hardness of the coating varies from 400 to 1000 HV. The Fe-B-$B_4$C-Si coating prepared with the highest laser beam scanning speed has the best corrosion resistance. Iosif Hulka et al. [98] reported the effects of laser cladding parameters on the morphology, wear resistance, and corrosion resistance of WC-Co/NiCrBSi composite coatings. They studied the effect of coating speed on the coating, performed hardness measurements and wear tests, and carried out corrosion tests in a 3.5% NaCl solution. The results show that the best coating speed affects the coatings' microstructure, composition, and hardness. The research results show that the coatings' sliding wear and corrosion properties can be effectively improved by optimizing the coating deposition rate. Meng et al. [99] analyzed a laser cladding TiC-Inconel 718 composite coating by statistical analyses and multi-objective process optimizations. The variance results showed that the scanning speed was the most significant factor affecting the coating's aspect ratio, dilution rate, and microhardness. The distance between primary dendrite arms increases with the increase of laser power and the decrease of scanning speed. On the contrary, the microhardness of the coating decreases. Then, the optimal coating can be obtained by the optimal method. The reasonable selection of laser cladding process parameters is very important to the cladding coating's geometric shape and mechanical properties.

Wu et al. [100] studied the microstructure and corrosion resistance of the laser cladding TiC-$Cr_7C_3$-CNTs metal-ceramic composite coating and found that the length of carbon nanotubes became shorter after laser cladding and the corrosion potential ($E_{corr}$) value had significant positive changes compared with the 304SS substrate. The decrease of the corrosion current ($I_{corr}$) indicates that the corrosion rate of the coating decreases. At the same time, the severe pitting problem observed in 304SS does not exist in the coating. Li et al. [101] found that the content of TiBCN would affect the microstructure and properties of Ti/TiBCN composite coatings. When the content of TiBCN was 60 wt.%, the coating was mainly composed of dendritic or rod-shaped TiBCN, a small amount of white bulk TiC, fine-coated TiN, a small amount of white dispersed bulk $TiB_2$, short whisker $Al_3Ti$, and TiAl (as shown in Figure 5). The maximum hardness of the composite coating is 1596 HV, which is about 4.6 times that of the substrate (as shown in Figure 6a). The corrosion potential ($E_{corr}$) and corrosion current ($I_{corr}$) of the composite coating are 1.258 V and $4.035 \times 10^{-5}$ A/cm$^2$, which are one order of magnitude lower than the corrosion current of the substrate ($1.172 \times 10^{-4}$ A/cm$^2$) (as shown in Figure 6b). The friction coefficient of the coating increases with the increase of Ti content (as shown in Figure 7). The wear mass loss of the substrate is 6.71 g, while the wear mass loss of the composite coating (1.22 g) is only 9/50 of that of the substrate (as shown in Figure 8). Zhao et al. [102] studied the microstructure and wear resistance of (TiC, TiN, and $B_4$C-reinforced) nickel-based coatings. The ceramic reinforced coating has an average microhardness of 766.8 HV$_{0.5}$. When 10% TiC, TiN, and $B_4$C were added to the initial Ni204 powder, the friction coefficient of ceramic reinforced coating was 0.47. Note that there are several phases, such as (Ti, Mo, Nb) (C, B,

N), carbides, nitrides, (Ti, Mo, Nb) (C, B, N) and (Ti, Mo, Nb) (C, N) ring phases around TiC and TiN, and Ti(C, N) ceramic phases), that can increase the hardness of the coating and reduce the friction coefficient. The multiphase ceramic-reinforced particles improve the friction properties of the coating.

The research results show that metal-ceramic cladding materials not only improve the bonding ability between ceramic particles and metal substrates but also play an important role in improving the wear resistance, corrosion resistance, and high-temperature oxidation resistance of the metal substrate. Metal-ceramic materials have great development potential and application space in laser cladding manufacturing. However, the splendid microstructure and properties of the composite coatings are closely related to the laser cladding process parameters, the content, and the type of ceramic-reinforced particles. Therefore, the optimization of laser cladding process parameters, the selection of ceramic reinforcement particles, and the strengthening behavior of coatings are helpful in improving the laser composite effect of the steel surface.

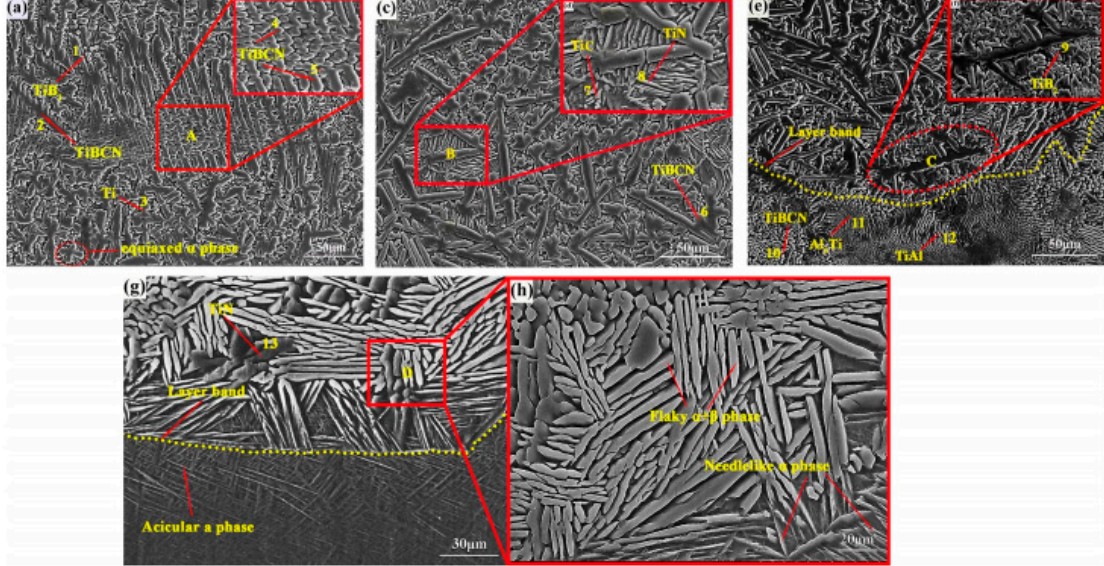

**Figure 5.** SEM images showing the microstructure of different zones of Ti/TiBCN composite coating with 60 wt.% TiBCN addition: (**a**) the coating region; (**c**) the upper part of the transition area; (**e**) the underpart of the transition area and the heat-affected zone; (**g**) the heat-affected zone and substrate; (**h**) the typical microstructure in the magnified image of the quadrangle [A], [B], [C], and [D] in (**a**,**c**,**e**,**g**), respectively.

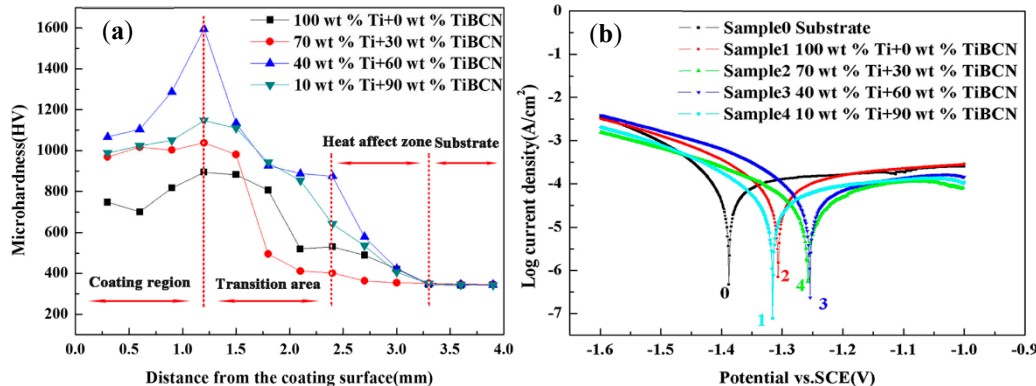

**Figure 6.** Microhardness and polarization curve of Ti/TiBCN composite coatings with the different TiBCN contents. (**a**) Microhardness, and (**b**) polarization curve.

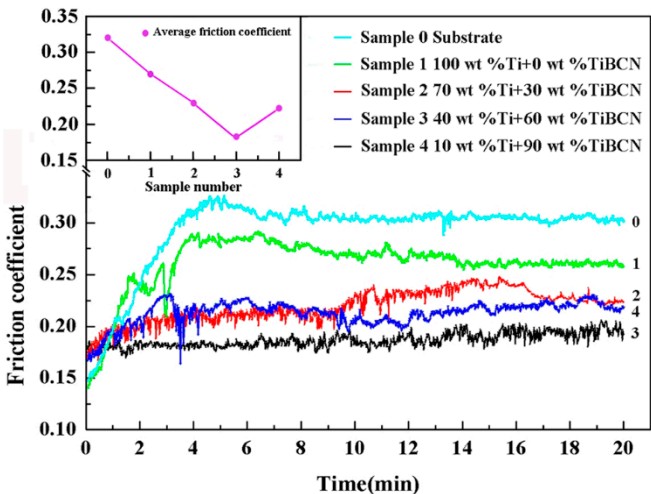

**Figure 7.** Friction coefficient of the substrate and Ti/TiBCN composite coatings with the different TiBCN contents.

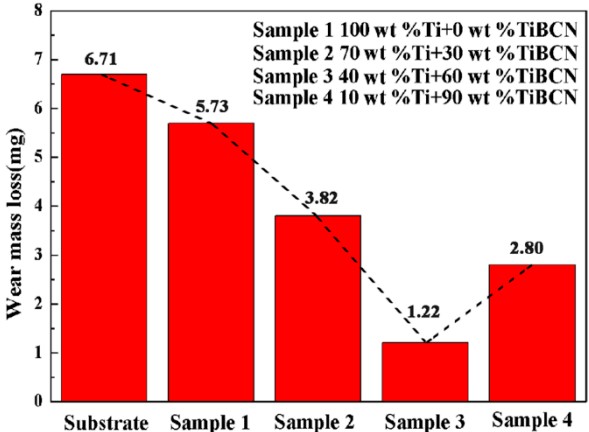

**Figure 8.** Wear mass loss of the substrate and Ti/TiBCN composite coatings with the different TiBCN contents.

Steel materials' laser cladding alloying and compounding process is extremely short and complex and involves many disciplines, such as materials, physics, and chemistry. The alloying and compounding results are affected by the cladding process and are closely related to the composition of cladding materials. The constitutive relationship between the laser cladding process, cladding materials, microstructure, and properties can be established, which can systematically analyze the alloying and composite process of laser cladding on steel and can also analyze and predict the cladding results to better improve the surface modification of steel.

## 5. Conclusions

By summarizing the research status of laser cladding process control and cladding materials of steel at home and abroad in recent years, the research progress of alloying and composites of steel and the existing problems were expounded. The analyses showed that:

(1) Laser cladding technology has been widely used in the alloying and compounding of steel, which provides a reliable technical choice for improving steel's surface properties and extending steel's service life.

(2) The current research on laser cladding alloying and compounding of steel includes many aspects, such as process control, cladding materials' design and selection, microstructure, and property characterization. According to the purpose of laser

cladding, the design of cladding materials matching the requirements deserves more attention.

(3) Laser cladding alloying and composite surface modification can improve steel materials' wear and corrosion resistance. Compared with laser cladding alloying, laser cladding can better improve the wear resistance of steel. In the future, the constitutive relationship between the laser cladding process, cladding materials, microstructure, and properties of steel should be further studied.

**Author Contributions:** T.H.: Conceptualization, methodology, writing—original, draft; K.Z.: investigation, methodology; Z.C.: formal analysis; Y.G.: investigation, resources, writing—review and editing. All authors have read and agreed to the published version of the manuscript.

**Funding:** This work was supported by the Project of Talent Introduction and Scientific Research Start-Up Fund of Nanjing Vocational University of Industry Technology #1, under Grant Number 201050622RS005, and the Jiangsu Province Industrial Sensing and Intelligent Manufacturing Equipment Engineering Research Center Open Fund Project #2, under Grant Number 201050622ZK002.

**Informed Consent Statement:** Not applicable.

**Data Availability Statement:** The data that support the findings of this study are available from the corresponding author, upon reasonable request.

**Conflicts of Interest:** The authors declare that they have no conflict of interest in this work.

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
