# Peer review of "Research Progress on Laser Cladding Alloying and Composite Processing of Steel Materials"

_metals, doi:10.3390/met12122055_

Round 1

Reviewer 1 Report (Previous Reviewer 2)

The review has been re-uploaded by the authors. Previously, I already got acquainted with the previous version of this review. The authors substantially revised and supplemented the review. The analysis now includes 99 previous studies. The review is well structured. The analysis of studies made in this review may be useful to other researchers. I think that in this version the review can be published in the scientific journal Metals. However, I have one remark. I made this remark during the last review and it has not been corrected. Here it is:

In figure 1, the dimensions are shown in "red" color, so they are hard to see. Change the color to "white".

Author Response

All concerns were addressed. Please see the uploaded response.

Reviewer 2 Report (New Reviewer)

The manuscript entitled “metals-2045494-Cladding” dealing with AM has been reviewed. The paper has been nicely written but needs significant improvement. Please follow my comments.

1.      Add a short note about the results to the abstract.

2.      Please improve the quality of  Figure 1 “Macroscopic structure of single channel cladding coating”.

3.      Change the ruller to the scale bar in figure 1 “Macroscopic structure of single channel cladding coating”.

4.      The explanation of Figure 2 “Influence of laser cladding process” needs more improvement.

5.      How authors selected the process parameters for the experimentation. This is important for repeating the test.

6.      The text has some typos. Please check them.

7.      Cladding is a type of directed energy deposition (DED) and has usage in additive manufacturing. Add a short note to your introduction based on the usage of additive manufacturing and add the following three papers.

·       Comparison of properties at the interface of deposited IN625 and mixture of IN625 SS304L by laser directed energy deposition and SS304L substrate

·       Thermo-fluid flow behavior of the IN718 molten pool in the laser directed energy deposition process under magnetic field

·       Additive manufacturing of Ti-Al functionally graded material by laser based directed energy deposition

·       Sandwich structure printing of Ti-Ni-Ti by directed energy deposition

Author Response

All concerns were addressed. Please see the uploaded response.

Round 2

Reviewer 2 Report (New Reviewer)

The paper is ready to publish.

Author Response

Firstly, we deeply appreciate your time and effort in reviewing our manuscript. Based on your comments, we have made relevant modifications to the previous work. And we have asked native English speakers and a professional language organization (NewTransing, LLC) to double check the language in this manuscript. The revised parts of the paper are tracked. Thanks!

This manuscript is a resubmission of an earlier submission. The following is a list of the peer review reports and author responses from that submission.

Round 1

Reviewer 1 Report

This paper fails to reflect the evolution and the state-of-the-art of laser cladding because it is biased to recent Chinese publications and instead of reviewing the topic repeats results previously published by the authors. There are many excellent reviews of laser cladding published since more than 40 years and several books, so I do not see any need for this publication.

Reviewer 2 Report

The paper under review is devoted to a very topical topic namely laser cladding. Laser cladding is a relatively new process, but has already proven itself to be very economically feasible and promising. Therefore, scientific materials devoted to this topic can be useful to a large number of researchers. However, when reviewing the paper, I had several questions and comments to the authors.

1. The peer-reviewed paper is uploaded as "review" while the template says "article". What is an article or review?

2. According to the material presented, the paper is more like a review, but for a review it is very small and contains an insufficient amount of reference for a review. There should be about 100 references.

Here are some articles that you might find helpful:

https://doi.org/10.1007/978-981-16-8922-2_3

https://doi.org/10.1007/s11015-021-01104-1

https://doi.org/10.3390/machines8040079

https://doi.org/10.1142/S0218625X22300118

https://doi.org/10.1088/1757-899X/969/1/012108

https://doi.org/10.3390/ma13020461

https://doi.org/10.3390/ma15124198

3. In paper, in my opinion, it is necessary to give material on the methods and technologies of laser cladding. Indicate their features, disadvantages and advantages relative to each other.

4. Laser cladding can be carried out not only with powder, but also with wire. This needs to be said in the review. Technologies that use wire are economically more efficient and are now becoming popular and in demand.

5. In Figure 1, the dimensions are shown in "red" color, so they are hard to see. Change the color to "white".

6. The work is not very well structured and does not have a deep system analysis. As an idea, I can recommend the following structure:

1) Introduction: definition of laser cladding, applications, advantages, problems, etc.

2) Methods and technologies of laser cladding: modes, equipment, deposition trajectories, etc.

3) Materials used for laser surfacing: powder and wire; chemical composition, purpose, etc.

4) Microstructure and mechanical properties of laser coatings.

5) Operational properties of coatings and practical application of laser cladding.

7. The reference is given in such a way that it is very difficult or impossible to view these articles. DOI must be specified.

I recommend the authors to finalize this very interesting and useful article and publish it in the scientific journal Metals.

Reviewer 3 Report

This review appears quite skewed towards steels and doest not offer a comprehensive introduction to the field, and especially the challenges of laser cladding. Only very specific microstructural cases are presented, and others that are of poor significance are introduced (e.g. CNTs). A more comprehensive and insightful review is required for publication.